# Top-Theta Attention: Sparsifying Transformers by Compensated Thresholding

## Abstract

We present Top-Theta (Top-θ) Attention, a training-free method for sparsifying transformer attention during inference. Our key insight is that static, per-head thresholds can be calibrated to retain the desired constant number of significant elements per attention row. This approach enables content-based sparsity without retraining, and it remains robust across data domains. We further introduce compensation techniques to preserve accuracy under aggressive sparsification, establishing attention thresholding as a practical and principled alternative to top-k attention. We provide extensive evaluation on natural language processing tasks, showing that Top-θ achieves 3–10× reduction in $\boldsymbol{V}$-cache usage and up to 10× fewer attention elements during inference while degrading no more than 1% in accuracy.

## 1 Introduction

The transformer architecture has revolutionized natural language processing Vaswani et al. (2017) and computer vision Dosovitskiy (2020) by enabling models to capture complex dependencies through self-attention mechanisms effectively. However, despite its advantages, the attention mechanism suffers from quadratic time and linear memory complexity Keles et al. (2023). Furthermore, the commonly used key-value (KV) cache optimization increases the memory requirement linearly with sequence length during the generative decoding phase. As a result, cache size requirements often exceed the physical limits of available memory Ge et al. (2024b), and memory bandwidth becomes a major bottleneck.

Attention approximations Wang et al. (2024); Fuad & Chen (2023), such as sparsification, promise a solution to these challenges by reducing the number of computations and the amount of data moved, focusing only on the most relevant tokens in a sequence. Research on sparsification has been predominantly focused on either fixed-sparsity patterns, which assume that specific token locations in the sequence are always more important (e.g., the first and last tokens in the sequence), or the content-based sparsity patterns, which require evaluating the attention scores to decide which tokens are more important (e.g., Top-k attention by Gupta et al. (2021)).

Our work focuses on the more challenging, content-based sparsity patterns. To exploit the sparsity potential, we investigate the sparsification of the attention elements through pruning by *comparing to a threshold value*. We calibrate the thresholds to select a desired average number of $k$ important tokens in every attention row. We find that calibrating model-specific thresholds is sufficient to replace the top-k search over the attention elements. Once the important tokens have been quickly determined by thresholding, the remaining tokens can be excluded from participating in the softmax computation and in the multiplication by the $\boldsymbol{V}$-matrix, thus avoiding the need to load the corresponding $\boldsymbol{V}$-rows. Moreover, to preserve the high accuracy in the downstream task, we propose numerical compensation methods such as softmax denominator compensation and mean $\boldsymbol{V}$-row compensation.

**Contributions** This work is taming the potential of content-based sparsity for more compute- and memory-efficient attention with negligible accuracy degradation. Our fundamental finding is:

*Static thresholds can be calibrated for a given attention head and used to sparsify its attention matrices to approximately $k$ elements per row.*

From this fundamental finding, we derive the Top-θ attention method, unlocking the following advantages:

1. *Efficiency* – 3× to 10× fewer $V$-rows needed and 10× less attention elements needed for LLaMA2 and LLaMA3 models to achieve same accuracy.

2. *Tiling compatible*. Thresholding is a simple elementwise operation with no row dependency, making it applicable to tiled attention matrices required for high-performance kernels and distributed inference.

3. *No retraining*. Threshold calibration requires only a few hundred samples and no retraining.

4. *Distribution shift resilience*. Thresholds remain consistent across input domains, representing a fundamental model characteristic that requires only one-time calibration.

## 2 Background

### 2.1 Transformer Models

Modern Large Language Models (LLM) used for text generation primarily employ decoder-only transformer layers, noted for strong zero-shot generalization Wang et al. (2022) and widespread success in chatbots and productivity tools. These models operate in two phases: processing the entire input at once (prefill) and generating tokens sequentially (generative decoding). Our research aims to enhance the underlying self-attention mechanism of decoder-only transformers.

### 2.2 Self-Attention and Sparsity

Multi-head self-attention (MHA) is the first computational step of the transformer layer. The MHA receives as input a sequence of tokens $X \in \mathbb{R}^{n \times D}$ where $n$ is the sequence length and $D$ is the hidden dimension. Each of the heads processes $X$ in parallel by first multiplying it by 3 different trained matrices $W_Q, W_K, W_V \in \mathbb{R}^{D \times d}$ obtaining 3 matrices $Q, K, V \in \mathbb{R}^{n \times d}$ and adding a positional encoding (e.g., RoPE Su et al. (2024)) to them. Then, a *pre-softmax attention matrix $A$* is computed from matrices $Q$ and $K$ (1). Although $A$ matrix is often normalized by $\sqrt{d}$ for numerical stability and masked by a causality mask, we omit these 2 steps for simplicity.

$$A = QK^T \tag{1}$$

After that, each row of the $A$ matrix is normalized by the Softmax operation, yielding the *post-softmax attention matrix $S$* (2), which is then multiplied by $V$ (3).

$$S = \text{row\_softmax}(A) \triangleq \left[ \frac{e^{A_{ij}}}{\sum_{k=1}^{n} e^{A_{ik}}} \right]_{\substack{1 \leq i \leq n \\ 1 \leq j \leq n}} \tag{2}$$

$$P = SV \tag{3}$$

**Prefill phase**  Both pre-, and post-softmax matrices are of the shape $n \times n$, in which the element $(i, j)$ signifies the importance of token $j$ for token $i$. The pre-softmax attention $A$ has a range of all real numbers distributed normally, whereas the post-softmax $S$ has the range of $[0, 1]$ and its distribution resembles log-normal, with the majority of the values concentrated near 0. The attention elements in the initial layers exhibit a less skewed distribution (i.e., high entropy), whereas the following layers have a more concentrated distribution (i.e., low entropy) with a few rare high-attention values Vig & Belinkov (2019); Nahshan et al. (2024). In both $A$ and $S$, a lower attention value has a lower contribution to the further computation because $S$ is obtained through an order-preserving transformation of $A$, after which $S$ is multiplied by the matrix $V \in \mathbb{R}^{n \times d}$ resulting in the output matrix $P \in \mathbb{R}^{n \times d}$ (3). Therefore, small values in column $i$ in $S$ diminish the impact of row $i$ of the $V$ matrix. *This fundamental property of the attention elements allows ranking them according to their significance and pruning the least significant ones for sparsification.*

**Generative decoding phase and KV-cache**  MHA employs a well-established performance optimization called KV-cache Shi et al. (2024) which allows processing a single embedded token $x \in \mathbb{R}^D$, computing only the current token's $q, k, v \in \mathbb{R}^d$ vectors, while the complete $K, V$ matrices are loaded from the cache (avoiding recomputation) and the new $k, v$ vectors are appended to them. In this situation, the attention matrices simplify to vectors $a, s \in \mathbb{R}^n$ representing the attention of the currently generated token to all the previous tokens, and the $P$ matrix becomes a single token

embedding $p = sV \in \mathbb{R}^d$. Due to the large size of the KV-caches, especially as the sequence length grows longer, the computation of the self-attention during the generative decoding is heavily limited by memory bandwidth, dominated by loading all the $n$ rows of the $K$ and $V$ matrices, while only performing 1 multiply-add per value read from memory. *Due to the memory bottleneck, a reduction of memory reads directly leads to a corresponding speed up.*

A variant of multi-head self-attention is called grouped query attention (GQA). The GQA introduces sharing a single pair of $K, V$ matrices for a group of $g$ heads (queries), which reduces the amount of $K, V$ data to load by a factor of $g$. Studies have shown that GQA has a marginal impact on the downstream task accuracy, making it a favorable optimization Ainslie et al. (2023).

### 2.3 Top-k Attention

One widely adopted approach to sparsifying the attention row is finding its top $k$ out of $n$ elements and discarding the rest, where $k$ is a hyperparameter Gupta et al. (2021). However, since the attention rows are often partitioned (tiling across the sequence dimension) and processed in parallel, computing the exact top-k values in a vector imposes an undesired full-row dependency, thereby constraining the tiling strategies. Moreover, the computation of the top elements requires several computational steps Zhang et al. (2023a), leading to a logarithmic complexity in the very best case. *We conjecture that the top-k search algorithms can be replaced by comparison to a carefully calibrated threshold.*

### 2.4 Motivating Example: Pipelined Kernel Constraints on Top-k Sparsification

Modern GPU attention kernels, such as paged-attention, serve as a strong example of how block-wise computation is used: each KV block undergoes $QK^T$, Softmax, and $SV$ computation in a sequential, pipelined manner Kwon et al. (2023). Within this structure, performing a global top-$k$ selection for each attention row is not feasible as the page-wise pipeline would need to materialize all partial results and synchronize globally, defeating the advantage of pipelining. By using elementwise thresholding instead, pruning decisions are made locally within each block and at each pipeline stage, maintaining compatibility with high-throughput parallel execution and avoiding full-row dependencies. This architectural alignment represents a fundamental advantage of threshold-based sparsification over top-$k$ approaches in realistic, high-performance transformer inference scenarios.

## 3 Top-θ Method

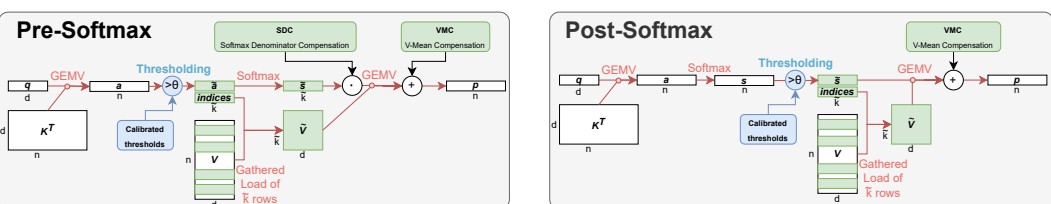

Figure 1: Two variants of Top-θ attention for inference at generative decoding.

Our proposed Top-θ attention involves comparing each attention vector against a calibrated threshold. Attention elements that fall below the threshold are pruned away from subsequent computations, enhancing the efficiency and focus of the model. Our underlying assumption is that a particular distribution of values characterizes each row of the attention matrix; therefore, using a static threshold should keep approximately the desired number of selected attention elements. Our motivation for using a threshold-based method instead of computing the top-k attention values is that thresholding is a simple elementwise operation, requiring only a constant time and involving no row dependency. In contrast, ranking the elements of a vector requires at least logarithmic time and depends on the full length of the vector. Top-θ can be seen as an approximate Top-k, as it allows calibrating the thresholds for specific user-defined $k$.

### 3.1 Threshold Calibration

Prior to using Top-θ for efficient inference, its thresholds need to be calibrated with respect to a user-defined parameter $k$. To this end, we present Algorithm 1 that calibrates a threshold $\theta_r(k) \in \mathbb{R}$

for attention row id $r$ of a given transformer layer and head. Calibration is performed by collecting the $(\frac{r-k}{k})$-quantile of each attention row from a set of calibration samples (l. 7), then averaging these per-sample thresholds to obtain $\theta_r(k)$ (l.19). This algorithm has to be performed for all layers and heads in parallel, and can be integrated into a single forward pass of the model (then disabled after calibration is complete). We found that using a single $k$ per layer suffices for robust performance, though different $k$ can be chosen per head or row.

For practical calibration, we recommend a calibration set of a few hundred samples, as we found that increasing the calibration set size mainly improves the fidelity of thresholding to keep the desired $k$ elements per row, but does not benefit the accuracy of the downstream task (see Appendix D). An optional offset hyperparameter $\alpha$ allows conservative adjustment of the final threshold. During calibration, a "Top-k at calibration" step (l. 8,15) ensures that subsequent layers' activations reflect the sparsification pattern, improving threshold stability at test time. Algorithm 1 in its presented form is calibrating pre-softmax thresholds, whereas it can be adapted to post-softmax by taking the quantile on the post-softmax attention $S, s$ rather than on the $A, a$. We note that calibrating for rows $r < k$ is unnecessary for the first $k$ rows due to causal masking.

Once the calibration is complete, the calibrated thresholds can be stored along with the model parameters, as they take negligible memory. For example, when 1200 per-row float16 thresholds are calibrated (as in ARC-C) for every head, the total memory for thresholds for an entire LLaMA-3-70B model (80 layers, 64 heads) is only $80 \cdot 64 \cdot 1200 \cdot 2 = 11.8$ Mbytes (0.0008% of the model size).

---

**Algorithm 1** Calibrate($\mathcal{C}, k, \alpha$) - 1 head threshold

**Require:** $\mathcal{C}$ (Calibration set of inputs)
**Require:** $k \in \mathbb{N}$ (elements to keep per attention row)
**Require:** $\alpha \in \mathbb{R}$ (calibration offset in std_devs)
1: $\Theta_r = \emptyset, \forall r$ {empty sets of observed thresholds}
2: **for** $X \in \mathcal{C}$ **do**
3:    **if** is_prefill($X$) **then**
4:       $A = Q(X)K^T(X)$
5:       $n = \text{num\_rows}(A)$
6:       **for** $r = k$ to $n - 1$ **do**
7:          $\Theta_r = \Theta_r \cup \{\text{quantile}_{\frac{n-k}{n}}(A_r)\}$
8:          $A_r = \text{top}_k(A_r)$
9:       **end for**
10:      $S = \text{row\_softmax}(A)$
11:    **else** {generative decoding, $X \in \mathbb{R}^d$}
12:       $a = Q(X)K^T(X)$
13:       $n = \text{length}(a)$
14:       $\Theta_n = \Theta_n \cup \{\text{quantile}_{\frac{n-k}{n}}(a)\}$
15:       $a = \text{top}_k(a)$
16:       $s = \text{softmax}(a)$
17:    **end if**
18: **end for**
19: **return** $\theta_r = \text{mean}(\Theta_r) + \alpha \cdot \text{std\_dev}(\Theta_r), \forall r$

---

Figure 2 visualizes two threshold variants of the 11$^{\text{th}}$ layer of LLaMA2-7b, which were calibrated for $k = 64$: pre-softmax, and post-softmax. Notably, the threshold depends both on the head and the sequence length (attention row id), justifying the calibration of an individual threshold for each attention (head, row) in a layer. Secondly, thresholds obtained pre-softmax approach a constant value as the sequence length increases, whereas thresholds obtained from the post-softmax tend to decrease with longer sequence lengths. The latter is an effect of the softmax normalization that reduces the share of essential tokens from the total row

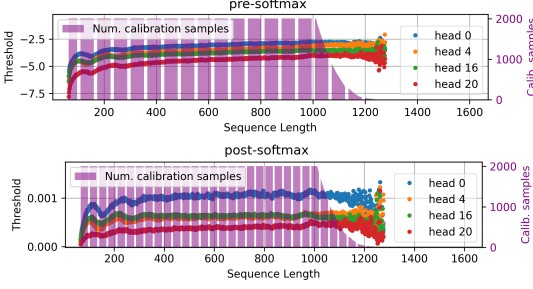

Figure 2: **Threshold values as a function of a sequence length** LLaMA2-7b, 11$^{\text{th}}$ transformer layer, calibration targeted $k = 64$.

sum of 1 as the sequence length increases; therefore, keeping them requires lowering the threshold. Scattering of the thresholds in the longer calibrated sequence lengths is due to the scarcity of the calibration samples in the dataset that corresponded to these sequence lengths (low purple bars in Figure 2). Appendix E presents a visualization of thresholds from additional layers.

**Further Optional Capabilities** While our method calibrates static thresholds per head and row (sequence length), an elegant extension would be fitting a parametric function to store thresholds as a function of row, reducing storage of the calibrated threshold and smoothing out the noisy tail at long-sequence-length calibrated thresholds 2. Such a fitting can be obtained via a weighted least squares solution, with the weights being the number of calibration samples obtained for each sequence length. Additionally, supporting variable switching between different $k$ values across layers or even

within a sequence (multi-k calibration) could enable even finer control over sparsity and performance trade-offs. We present the multi-k calibration in Appendix F.

## 3.2 Top-θ Attention Inference

The Top-θ attention inference operation in Figure 1 focuses on a single transformer layer $l$, single head $h$, and single attention row $n$ of length $n$ (as in generative decoding). The attention vector is compared against the calibrated threshold value $\theta_n$. Attention elements that do not pass the threshold are discarded. Instead of multiplying by the entire $V$ matrix, only the selected $\tilde{k}$ row indices are loaded to a compact matrix $\tilde{V}$, which is used to compute the final product $p$.

Technical details: (i) Since the calibration set might not have covered the entire range of sequence lengths $r \in \{k, k+1, \ldots\}$, at inference, the threshold of the nearest calibrated sequence length is used. (ii) Threshold calibrated for a target $k$ is guaranteed to select $k$ attention elements per row only on average, i.e., actual counts ($\tilde{k}$) may slightly vary due to input-dependent attention distributions. To handle cases where $\tilde{k} > k$, we experimented with capping the selection at $k$ elements, prioritizing first and last tokens, but found that this noticeably degraded performance on generative tasks. Instead, we mitigated the variability of $\tilde{k}$ by increasing the calibration set size (Appendix D).

The rest of this section introduces two efficient compensation mechanisms that mitigate accuracy degradation in Top-θ attention through improved mathematical approximation of full self-attention.

### 3.2.1 Softmax Denominator Compensation (SDC)

Since the post-softmax sparsification showed higher accuracies in our experiments compared to pre-softmax, we are interested in approximating post-softmax-sparsified attention ($\tilde{s}$) using pre-softmax-sparsified attention ($\tilde{a}$). Let $\mathcal{I} \subseteq \{0, \ldots, n-1\}$ denote a set of indices that we intend to keep during the sparsification, and let us establish the relation between post-softmax sparsified vector $\tilde{s}$ and the pre-softmax sparsified vector that underwent softmax ($\mathrm{softmax}(\tilde{a})$). Let $R$ and $E$ denote the sums of exponents of the selected and discarded elements, respectively.

$$\forall i \in \mathcal{I}, \tilde{s}_i = \mathrm{softmax}(a)_i = \frac{e^{a_i - \max_l a_l}}{\sum_{j=0}^{n-1} e^{a_j - \max_l a_l}} = \frac{e^{a_i - \max_l a_l}}{\sum_{j \in \mathcal{I}} e^{a_j - \max_l a_l} + \sum_{j \notin \mathcal{I}} e^{a_j - \max_l a_l}}$$

$$= \frac{e^{a_i - \max_l a_l}}{R + E} = \frac{e^{a_i - \max_l a_l}}{R} \cdot \frac{R}{R + E} = \mathrm{softmax}(\tilde{a})_i \cdot \frac{R}{R + E} \quad (4)$$

In other words, to achieve a post-softmax sparsification effect, one can perform pre-softmax sparsification, estimate $\tilde{E} \approx E$, and compensate by multiplying by a factor of $R/(R + \tilde{E})$. The multiplication step can be applied after the softmax (2) or even after the $SV$ product (3), similarly to the flash-attention Dao et al. (2022). We consider 3 estimations:

1. `offline-calibrated`: calibrate a static compensation value $\tilde{E}$ for every (layer, head, row) similarly to the method of threshold calibration. Cost: 1 scalar load per head, extra memory is required (equal to the thresholds storage), and extra calibration is required.

2. `exp-threshold`: $\tilde{E} = \gamma(n - \tilde{k})e^{\theta}$ where $\theta$ is the calibrated threshold for the current sequence length $n$, and $\tilde{k}$ is the number of selected attention elements. The intuition behind this approximation is that all the not-selected attention elements are less than $\theta$. The $\gamma$ is a small constant hyperparameter that we set to $0.05$ to approximate the difference between the sum of exponentiated thresholds and actual exponentiated discarded attention elements. Cost: 3 scalar operations per attention head, no extra memory, no extra calibration required.

3. `exact`: compute $\tilde{E} = E$ by summing up exponents of the non-selected row elements. Cost: $2(n - \tilde{k})$ scalar operations per attention head, no extra memory, no extra calibration required.

Assuming that $\mathrm{argmax}_l a_l$ is included in the selected elements $\mathcal{I}$, we do not correct the maximum value subtracted from the exponents during softmax computation.

### 3.2.2 V-Mean-Compensation (VMC)

Applying Top-θ in the following two cases will result in attention vector rows not summing up to 1: (i) post-softmax, (ii) pre-softmax followed by SDC. As a result, the value given by the product of

non-selected attention elements and their corresponding $V$-rows will be missing in the final product (3). We denote these cases as "eligible for VMC" and we compensate for the missing value in (3) by adding a mean row $\mu$ of the $V$ matrix scaled by the sum of all the discarded attention elements $\beta$. Equation 5 lists the computation of the corresponding $V$-mean-compensated product $\hat{p} \approx sV$, where $\tilde{s} \in [0, 1]^{\tilde{k}}$ is a post-softmax attention vector that underwent Top-$\theta$ (containing the selected $\tilde{k}$ out of $n$ attention elements) and is eligible for VMC, and $\tilde{V}$ contains selected $\tilde{k}$ rows of $V$.

$$\mu = \frac{1}{n}\sum_{i=0}^{n-1} V_i, \qquad \beta = 1 - \sum_{i=0}^{\tilde{k}-1} \tilde{s}_i, \qquad \hat{p} = \tilde{s}\tilde{V} + \beta\mu \tag{5}$$

We formally justify the VMC approximation in Appendix G. The intuition behind it is to approximate each of the $n - \tilde{k}$ discarded values by an average and to multiply it by an average row of the $V$ matrix. Such compensation can improve as the sequence length $n$ increases because the number of averaged elements in $\mu$ and $\beta$ will increase accordingly. During generative decoding, $\mu$ can be maintained as a running mean to avoid full recalculation.

## 4 EVALUATIONS

We use the LM Evaluation Harness Gao et al. (2024) to evaluate the normalized accuracy metric on multiple-choice Q&A datasets, including their standard few-shot settings. In addition, we assess the Human-eval dataset using the official evaluation harness Chen et al. (2021) to measure the pass@1 metric on all 164 code generation tasks, and two long sequence summarization tasks of LongBench (qmsum and gov_report) using their official evaluation setup Bai et al. (2024). Our hardware setup required GPUs with a total of 192GB (to run 70B models), and a CPU with 300GB of physical memory. Reproducing the 7B and 8B model results on smaller datasets (i.e., not LongBench) can be accomplished using a single 24GB VRAM GPU. We provide our full source code with reproducibility scripts.

We evaluate 3 main attention variants: (i) Baseline – full attention, without any sparsification; (ii) Top-k – keep $k$ attention elements per row; (iii) Top-$\theta$ – keep $\tilde{k} \approx k$ attention elements by thresholding. We also apply our compensation methods to the Top-k baselines for a fair comparison.

### 4.1 TOP-$\theta$ OVERALL PERFORMANCE

In Figure 3, LLaMA2 Touvron et al. (2023) and LLaMA3 Grattafiori et al. (2024) models are evaluated on Q&A tasks, where the $k$ parameter of Top-k and of the Top-$\theta$ is swept from 32 to 512. The first two layers are kept at $k = 512$. The calibration set used for Top-$\theta$ in each dataset is $10\%$ of the training or validation sets (different from the test set). The $x$-axis shows the fraction of the attention elements that are involved in computation, normalized to an average number of attention elements in the entire model in a given forward pass. The $y$-axis shows the normalized accuracy. The main observation is that in all models, both Top-k and Top-$\theta$ have *increased* the accuracy (by $0.2\% - 1\%$) compared to the baseline while pruning away a significant portion of attention elements ($2\times - 5\times$ fewer elements were active). Second, post-softmax sparsification performs consistently better in both Top-k and Top-$\theta$, compared to the pre-softmax sparsification.

In Figure 4, we focus on the generative tasks, where the main bottleneck is the reading of the KV cache. To demonstrate domain adaptation capabilities of the calibrated Top-$\theta$ thresholds, they were first calibrated on a different dataset (ARC-C) and then loaded for Human-eval and LongBench evaluation. We examine LLaMA-3-Instruct and LLaMA-3.1-Instruct models, and we use 0 temperature for generation. In Figures 4a and 4b we focus on Human-eval task showing the tradeoff between the output quality (pass@1) and the number of required $V$-rows. The $k$ parameter was swept between 32 and 512 with the first two layers set at 512. The post-softmax Top-$\theta$ performs the best in both 8B and 70B models, preserving pass@1 within $1\%$ of the baseline while reducing the required $V$-rows by $3\times$. Impressively, in the 70B model, Top-$\theta$ offered a $5\times$ reduction at the expense of $1\%$ of the pass@1. On LongBench tasks (Figures 4c and 4d), where the average prompt lengths reached around 15k tokens, the $k$ parameter was swept in between 128 and 768 with the first two layers set at 768., and Top-$\theta$ showed even higher reductions of the required $V$-rows - up to $10\times$ despite the GQA. Such a reduction was achieved using thresholds calibrated for as few as $k = 128$ elements. The Rouge-L score (higher is better) stayed within $1\%$ of the baseline, often improved by $0.5\%$.

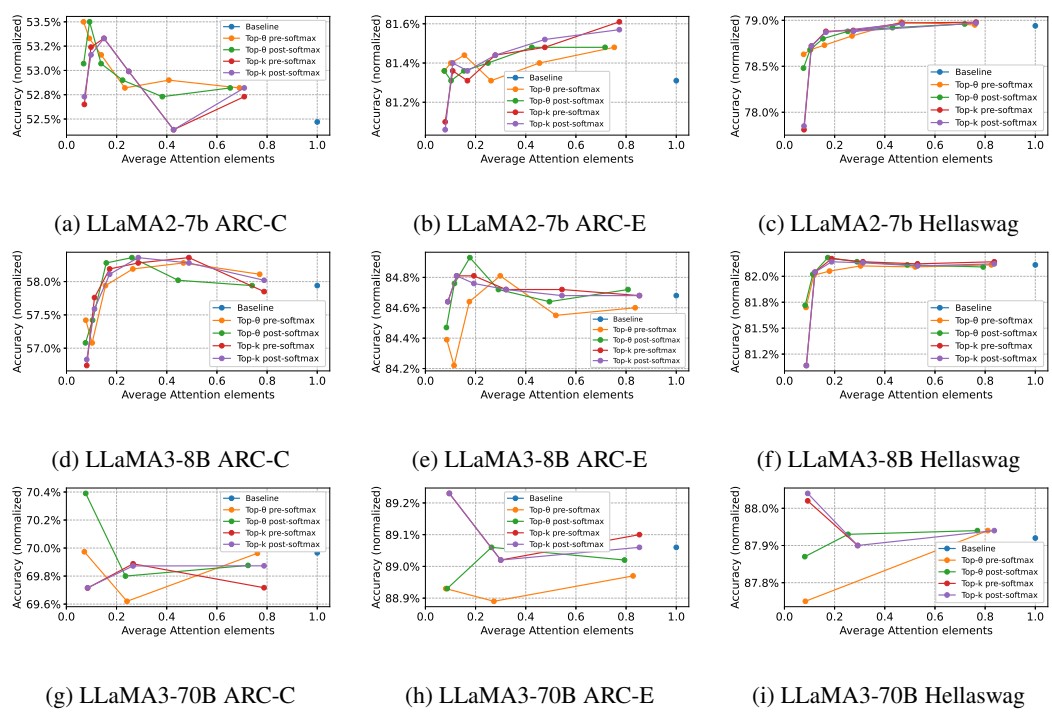

(a) LLaMA2-7b ARC-C     (b) LLaMA2-7b ARC-E     (c) LLaMA2-7b Hellaswag

(d) LLaMA3-8B ARC-C     (e) LLaMA3-8B ARC-E     (f) LLaMA3-8B Hellaswag

(g) LLaMA3-70B ARC-C     (h) LLaMA3-70B ARC-E     (i) LLaMA3-70B Hellaswag

Figure 3: **Prefill-based tasks** - Tradeoff between model accuracy (y-axis) and the portion of kept attention elements per attention head (x-axis). All post-softmax Top-k and Top-θ employ VMC, and all pre-softmax variants employ both VMC and exact SDC. These compensations achieve little, if any, accuracy degradation while achieving up to 10× reduction in the attention elements.

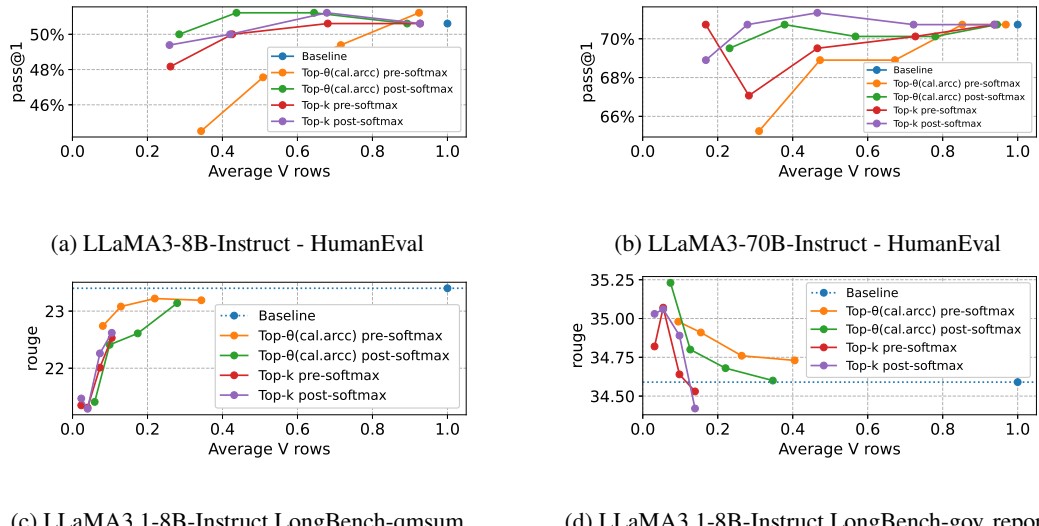

(a) LLaMA3-8B-Instruct - HumanEval     (b) LLaMA3-70B-Instruct - HumanEval

(c) LLaMA3.1-8B-Instruct LongBench-qmsum     (d) LLaMA3.1-8B-Instruct LongBench-gov_report

Figure 4: **Generative Tasks** – Tradeoff between model accuracy (y-axis) and the portion of required $V$-rows per group of heads (x-axis). The Top-θ variants employ a threshold calibrated on ARC-C dataset. All post-softmax Top-k and Top-θ employ VMC, and all pre-softmax variants employ VMC and exact SDC. 3× and 10× reduction of $V$ rows is achieved on Human-eval and LongBench, respectively.

Overall, both Top-θ and Top-k performed similarly well on both Q&A and generative tasks. For a version of Figures 3 and 4 with standard deviations of the collected metrics, the reader can refer to Appendix H. The rest of this section shows an ablation study of our method.

## 4.2 Pre- vs Post-Softmax Thresholding

We evaluate the impact of attention matrix sparsification in its two main variants: on matrix $\boldsymbol{A}$ and on the post-softmax matrix ($\boldsymbol{S}$), where each of these two variants requires individual calibration. Figure 5 depicts how the different thresholdings impact the accuracy of LLaMA2-7b on the Hellaswag dataset. For comparison, the Top-k approach is also evaluated alongside Top-$\theta$. We conclude that *post-softmax preserves more accuracy compared to pre-softmax thresholding*, and we provide extended results on additional datasets (see Appendix I).

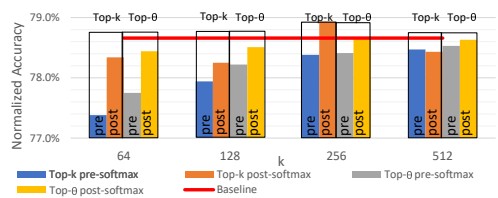

Figure 5: Pre- vs. post-softmax Top-k/$\theta$

## 4.3 Thresholding Different Layers

We explored the impact of thresholding different layers with a different target $k$. For the LLaMA models, we have observed that targeting a slightly higher $k$ in the first layers is crucial. As a best practice, we found keeping 2 initial layers at $k = 512$, whereas the rest of the layers could be sparsified more aggressively. Figure 6 shows the LLaMA2-7b model accuracy on Hellaswag, comparing Top-k and Top-$\theta$ using higher $k$ in first 2 layers against using equal $k$

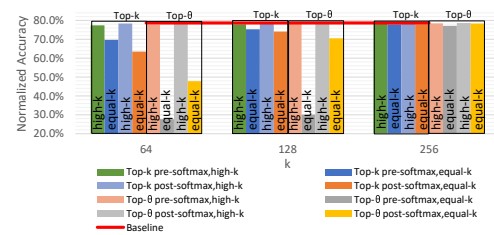

Figure 6: High-k in first two layers

in all layers. All variants do not perform any compensations. *The conclusion is that preserving denser attention matrices in the early layers is beneficial for the accuracy of a downstream task, which is aligned with some of the works on quantization* Tang et al. (2024); Huang et al. (2024).

## 4.4 Numerical Compensations

We evaluate the proposed numerical compensation methods SDC and VMC, finding that more explicit SDC variants (exp-threshold, exact) substantially recover degraded accuracy on the challenging Hellaswag task with LLaMA2-7b and Top-$\theta$ attention, as shown in Figure 7; combining SDC with VMC further improves results, effectively closing the accuracy gap between the baseline and the non-compensated Top-$\theta$ attention. However, on the ARC-C and ARC-E datasets, Top-$\theta$ attention alone already outperforms

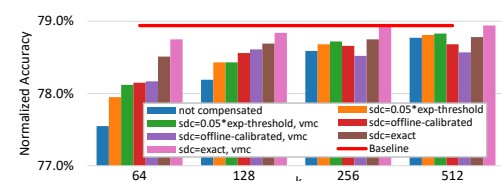

Figure 7: **SDC and VMC compensations** impact the accuracy positively.

the baseline by reducing attention noise, so applying compensations in these cases offers little benefit and may even reduce accuracy back to baseline levels by reintroducing noise. *Overall, SDC and VMC compensations almost entirely recover the accuracy.*

## 4.5 Impact of Grouped Query Attention (GQA)

In GQA, multiple attention heads share the same $\boldsymbol{V}$ matrix; for LLaMA-3-8B models with a group size of $g = 4$, this means up to $4k$ $\boldsymbol{V}$ rows could be needed if heads select completely different tokens. However, as shown in Figure 8, both Top-k and Top-$\theta$ with $k = 128$ typically select only 250–300 $\boldsymbol{V}$ rows per group, indicating substantial agreement among heads - a pattern observed across many layers and further illustrated in Appendix L.

While our method uses independent per-head selections, GQA sparsification could be improved by

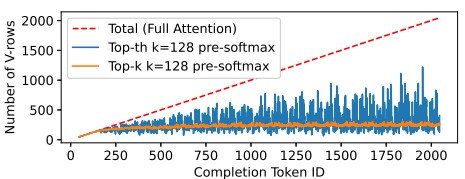

Figure 8: **GQA impact** - Number of required $\boldsymbol{V}$-rows for every generated token, LLaMA-3-8B-Instruct (layer 2, first GQA group), Humaneval task #25.

considering two alternatives: (1) discarding $\boldsymbol{V}$-rows requested by few heads, or (2) augmenting selections with the group's union. Discarding (1) maintains a stricter budget but adds complexity (e.g., vote thresholds) and risks dropping crucial tokens. Conversely, augmenting (2) is simpler

to implement via a mask union. Limited evaluation showed only a minor accuracy uplift using augmenting (2), which we attribute to the added tokens being less important to the individual heads.

## 4.6 DISTRIBUTION SHIFTS

To examine how domain-sensitive the calibrated threshold is, we evaluate on Human-eval the two following Top-θ variants: 1) calibrated on Q&A dataset of ARC-C, and 2) calibrated on the first 10% of Human-eval tasks. As seen in Figure 9a, the Top-θ post-softmax is even more accurate when using thresholds calibrated on a different dataset. For pre-softmax, there is a benefit of calibrating on the same dataset. We also evaluate using MedMCQA, where we compare the Top-θ calibrated on ARC-C with Top-k on Figure 9b - showing that both sparsification methods perform equally well. *Overall, the thresholds show resilience towards distribution shift, suggesting that they are strongly associated with the model rather than with the data. This allows calibrating thresholds once per model.*

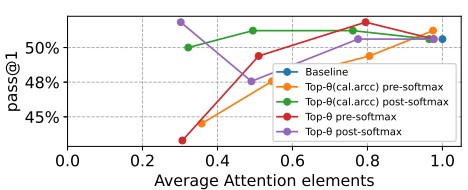

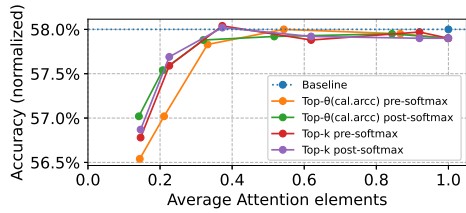

(a) LLaMA3-8B-Instruct - Human-eval

(b) LLaMA3-8B MedMCQA

Figure 9: **Distribution shift** - Top-θ calibrated on different task (labeled with *cal.arcc*) shows comparable accuracy and attention reduction compared to Top-θ calibrated on the same task 9a, and compared to Top-k 9b.

## 4.7 TOP-THETA KERNEL

We have implemented a prototype of the Top-θ MHSA attention kernel on Ascend NPU (containing 24 Davinci cores, 376 TOPS/sec FP16 computational throughput, 1.8TB/sec global memory bandwidth) and observed a kernel speedup of $1.17\times$ compared to an optimized decoding flash attention kernel `IncrementalFlashAttentionV4` Huawei (2024). The setup was 48 heads, $D = 128$, Top-θ target $k = 128$ out of the sequence length of $n = 2048$ tokens. Importantly, performance gains from sparse kernels are highly hardware-dependent, as they rely on low-level specifications such as DMA, bandwidth, and parallel execution capabilities. In our setup, scattered reads of selected $V$-rows incurred notable per-transaction overhead compared to the contiguous memory access of dense kernels, echoing observations on DRAM and HBM-based systems Miao et al. (2019); Chatterjee et al. (2014). These findings highlight the need for gather-efficient memory subsystems and compact KV-cache algorithms in future architectures.

## 5 RELATED WORK

**Eviction-based** Early sparse transformers enforced static sparsity patterns to select important tokens. *StreamingLLM* Xiao et al. (2024b) retains in its KV cache only "attention sinks" and a fixed local window of tokens, while *Swin Transformer* Liu et al. (2021) restricts attention to local patches. Though computationally efficient, these patterns fail to capture dynamic, long-range dependencies. Combining fixed patterns with profiling, *DuoAttention* Xiao et al. (2024a) classifies attention heads into "retrieval" (full context) and "streaming" (local/sink only), pruning the KV cache only for the retrieval heads. In contrast, Top-θ dynamically adapts the attention pattern at runtime per layer, head, and sequence length granularity.

The subsequent sparse transformers shifted more toward a content-dependent sparsity, tracking the generated tokens' attention instead of applying static rules to them. $H_2O$ Zhang et al. (2023b) uses a Heavy-Hitter Oracle token selector based on accumulated token-scores. *SnapKV* Li et al. (2024) reduces the cache to a predefined size budget by performing a scoring step at prefill. The *PyramidKV* Cai et al. (2025) builds upon it by adding a layer-wise retention strategy. *FastGen* Ge et al. (2024a) profiles attention structure at prefill to construct adaptive KV caches.

Importantly, both the fixed and content-based strategies described above operate as *eviction-based* mechanisms: they permanently remove selected KV entries, thereby discarding information to reduce memory footprint and bandwidth usage. In contrast, Top-θ departs from the objective of minimizing

Table 1: **Comparison of Top-θ with state-of-the-art sparse training-free attention methods**. **Sparsity Pattern** indicates if the selection is fixed, fully content-based, or a hybrid of the two. **Calibration** denotes if a pre-inference pass is required. **Prefill Cost** indicates whether a significant computation at prefill is required (e.g. initial token scoring or metadata creation). **Decoding Cost** reflects the overhead at generative decoding (e.g., Sorting vs. Thresholding). **Eviction** indicates if the method permanently drops tokens from the cache (capacity reduction) vs. bandwidth reduction only. **Compensated** indicates if the method corrects for excluded values to preserve accuracy.

| Method | Sparsity Pattern | Calibration | Prefill Cost | Decoding Cost | Eviction | Compensated |
|---|---|---|---|---|---|---|
| **StreamingLLM** Xiao et al. (2024b) | Fixed (Sink+Local) | No | No | Low (Mask) | Yes | No |
| **DuoAttention** Xiao et al. (2024a) | Fixed (Head-wise) | Yes | No | Low (Mask) | Yes | No |
| **H₂O** Zhang et al. (2023b) | Content-Based | No | Yes | High (Sort) | Yes | No |
| **SnapKV** Cai et al. (2025) | Content-Based | No | Yes | Low (Static) | Yes | No |
| **PyramidKV** Cai et al. (2025) | Hybrid (Layer-wise) | No | Yes | Low (Static) | Yes | No |
| **FastGen** Ge et al. (2024a) | Content-Based | No | Yes | Low (Lookup) | Yes | No |
| **Top-$k$ Attention** Gupta et al. (2021) | Content-Based | No | No | High (Sort) | No | No |
| **SparQ** Ribar et al. (2023) | Content-Based | No | No | High (Iterative Approx. + Sort) | No | Yes |
| **Quest** Tang et al. (2024) | Content-Based | No | Yes | Medium (Approx + small topk) | No | No |
| **L-Serve** Yang et al. (2025) | Hybrid(Duo+Quest) | Yes | Yes | High (Mask+Selector) | Yes | No |
| **Top-θ (Ours)** | **Content-Based** | **Yes** | **No** | **Low (Element-wise)** | **No** | **Yes** |

KV size and instead focuses exclusively on reducing memory bandwidth, enabling it to preserve accuracy by retaining all KV information. This design allows the model to repeatedly attend to tokens that would have been irreversibly discarded by the aforementioned eviction methods.

**Dynamic and Query-Aware Sparsity.** More advanced methods estimate token criticality dynamically at runtime. *Quest* Tang et al. (2024) approximates attention scores using min/max values of cached keys to select critical KV pages. *L-Serve* Yang et al. (2025) optimizes serving with profiling and tiered dynamic sparsity. *ShadowKV* Sun et al. (2024) offloads the Value cache and uses a low-rank Key cache for scoring. While effective, these approaches introduce complex system overheads (e.g., offloading, scoring). Top-θ offers a simpler, lightweight alternative requiring no initialization at prefill and no complex approximations at decoding, achieving dynamic sparsity via hardware-friendly thresholding without system-level offloading or complex estimation.

**Approximation via Top-$k$ and Thresholding.** Top-k attention Gupta et al. (2021) is the most direct content-based method, keeping the $k$ largest scores. However, efficient Top-k selection is hardware-unfriendly due to sorting overheads (Section 2.3) and constraints on parallelization (Section 2.4). *SparQ* Ribar et al. (2023) improves accuracy via value-mean compensation but retains the sorting bottleneck. Unlike these methods, Top-θ avoids the sorting cost. It employs a calibrated threshold $\theta$ for element-wise pruning, achieving $O(N)$ efficiency. Furthermore, Top-θ integrates a compensation mechanism, ensuring high approximation accuracy while benefiting from the fast threshold check.

## 6 LIMITATIONS

This work focuses on sparsifying attention and reducing value-matrix accesses with an emphasis on accuracy preservation. We do not extensively evaluate wall-clock runtime improvements, as achieving practical speedups requires specialized, hardware-aware kernel optimizations beyond the scope of this paper. Our experiments are limited to decoder-only transformer models within the LLaMA family, which cover common attention mechanisms such as GQA and MHSA that span across some of the most successful models (Mistral, QWEN, Phi, etc.). Extending the approach to larger or different transformer architectures and other attention methods remains an important direction for future work. Additionally, factors such as model size, sequence length, and attention patterns may influence the trade-offs between sparsity and accuracy in other settings.

## 7 CONCLUSION

We have presented Top-θ, a new sparse attention algorithm based on fixed and calibrated thresholds. At inference time, it boils down to a simple elementwise operation, which overcomes the limitation of full-row dependent algorithms, such as Top-k, and thereby unconstrains tiling and distributed inference implementations. Moreover, only a small subset of calibration samples is required to calibrate the thresholds, which we showed to be resilient to distribution shifts. Therefore, a short calibration is needed once per model. Furthermore, we show minor to no performance degradation using 10× less attention elements at the computationally-bound prefill phase and using 3× (for shorter sequence tasks) to 10× less $V$ matrix rows (for longer sequence tasks) at the memory bandwidth-bound generative decoding phase. These reductions unlock promising speedups for inference.

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

## A    APPENDIX

## B    IMPACT STATEMENT

This paper presents work whose goal is to advance the field of efficient Machine Learning. All potential societal consequences are mostly unrelated to the specific work but are more related to machine learning applications in general. One potential positive consequence of our work is that LLM technologies can be adopted by vendors and systems with lower resources, since our proposal unlocks deployment of LLMs in systems with lower memory bandwidth. We clearly and transparently state that our method is a lossy method (i.e., the accuracy of the downstream LLM task may degrade in favor of higher performance). Lossy deep learning models are a known practice in literature (e.g., when applying quantization and pruning), hence one should employ sufficient guardrails when deploying a lossy method in security-sensitive scenarios.

## C    LLM USAGE

In preparing this manuscript, we used large language models (LLMs) solely as a tool to aid and polish the writing. Namely, the LLM assistance was limited to improving language clarity and grammar in a few isolated sentences. All content, ideas, and research contributions presented in this paper are the original work of the authors. We take full responsibility for the accuracy, validity, and integrity of the paper. No research ideation, experimental design, or content generation was delegated to LLMs.

## D  THRESHOLD CALIBRATION SET SIZE

We have experimented with various calibration set sizes, and the main finding was that even with as few as 8 calibration samples, the model retains good accuracy. As the calibration set size grows larger, to a few hundred, the number of attention elements selected via thresholding ($\tilde{k}$) is approaching the desired $k$ on average, with a smaller variance. See Figure 10 where we plot on the left the ratio between the effective number of elements that passed the threshold ($\tilde{k}$) and the desired $k$ (the closer to 1.0 the better).

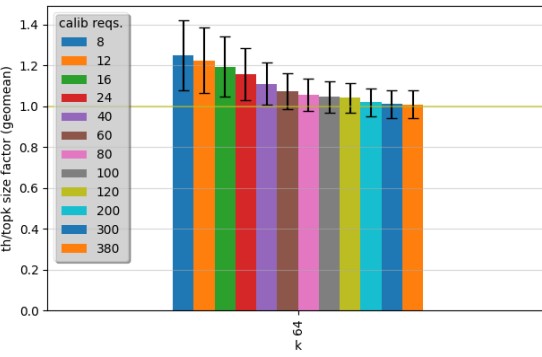 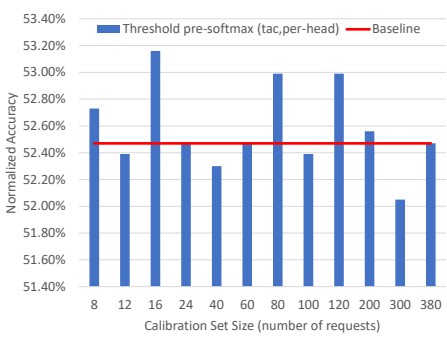

Figure 10: **Calibration set size impact** - LLaMA2-7B, calibration and evaluation on ARC-C. **Left**: the Y axis shows the average ratio $\tilde{k}/k$ (the closer to 1 the better the approximation of topk) and the X-axis (different bars) refers to different calibrations that were performed with calibration set sizes ranging from 8 to 380. **Right**: accuracy of LLaMA2-7b with Top-$\theta$ on ARC-C dataset as a function of the calibration set size used to calibrate the thresholds.

## E  THRESHOLD CALIBRATION

In this appendix section, we show more calibrated threshold values (as a function of the sequence length) in more layers and heads. Figure 11 visualizes the thresholds as a function of a sequence length that was calibrated for The LLaMA2-7b model for the Hellaswag dataset. Different heads have different threshold values, hence the importance of per-head individual calibrations.

## F  MULTI-K CUMULATIVE CALIBRATION

Algorithm 1, which we presented in the paper, calibrates thresholds of a single attention head for a single target $k$ value. We propose to generalize it to calibrate all thresholds for a broader range of target $k$ and to do it in one pass over the calibration samples. We call such a calibration procedure "Multi-k Cumulative" (MKC).

The goal of MKC calibration is to construct a multi-$k$ threshold function $\theta(k) : \mathbb{N} \to \mathbb{R}$ for every (layer $l$, head $h$, attention row id $r$). The user will be able to query such a function using their $k$ of choice and obtain the threshold needed to accommodate this $k$. Such an ability allows a very flexible control over the sparsification of the model, which is a highly desirable tool for LLM inference service that should be able to dynamically trade some accuracy for speedup.

**MKC algorithm**   Algorithm 2 describes how a threshold function $\theta(k)$ can be calibrated. First, for every given calibration sample, let $v$ represent the $r^{th}$ attention row. The $v$ undergoes a sorting and then is treated as a sequence of half-open intervals. For each interval, we treat its smaller endpoint as its threshold, and we also associate with it an effective $k$ (effective $k$ of a threshold w.r.t the vector $v$ is the number of elements that are greater than the threshold). For example, on line 2 we represent by $\langle r-1, [v_1, v_2) \rangle$ the interval between $v_1$ (inclusive) and $v_2$ (exclusive) that corresponds to the threshold $v_1$ and an effective $k$ of $r-1$. Second, per-calibration-sample interval sequences collected in the set $\Theta_r$ are merged to represent the function $\bar{k}(\theta) : \mathbb{R} \to \mathbb{R}$ which maps each interval

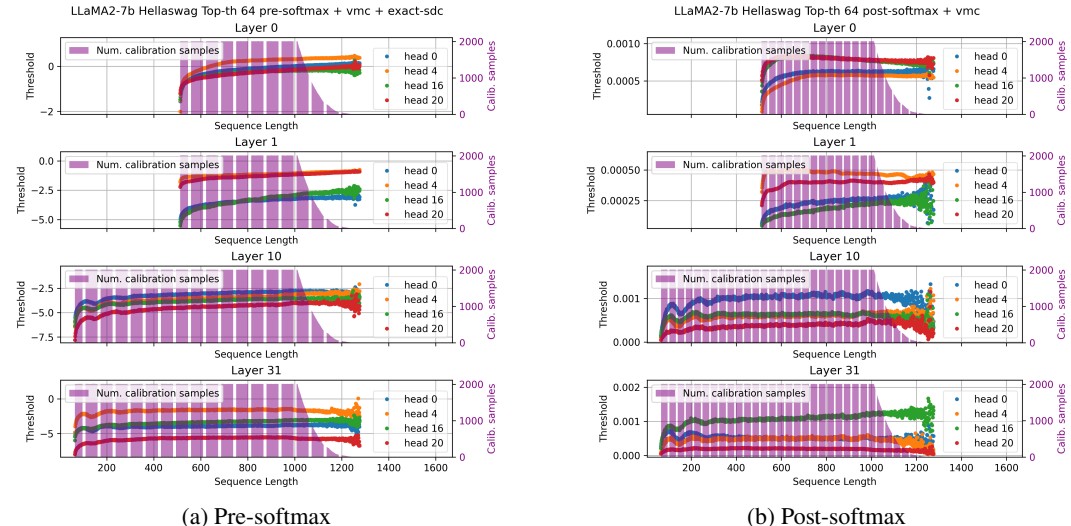

(a) Pre-softmax (b) Post-softmax

Figure 11: **Threshold values as a function of a sequence length in LLaMA2-7b**. Left Column: pre-softmax, Right Column: post-softmax. Scatter plots: final calibrated threshold values. Rows top to bottom: layer 0,1,10,31. Purple columns: number of calibration inputs that contained the respective sequence length. Layers 0 and 1 were calibrated for $k = 512$, and layers 10 and 31 were calibrated for $k = 64$.

to a threshold and to an average effective $k$ achievable by this threshold on the calibration set. This merging is done by the **MergeIntervals**($\Theta_r$) subroutine, which computes the following:

$$\bar{k}_r(\theta) = \frac{1}{|\Theta_{r,\theta}|} \sum t_0, \forall \theta \tag{6}$$

where we define $\Theta_{r,\theta} = \{t | \theta \in t_1 \wedge t \in seq \wedge seq \in \Theta_r\}$ as a subset of intervals that include the $\theta$ in the interval. Note that the resulting average effective $k$ might be fractional due to the averaging. Finally, the desired $\theta(k)$ is given by the inverse of $\bar{k}(\theta)$, at points where the function $\bar{k}(\theta)$ maps to a natural value. The full implementation of the MergeIntervals routine is available in the supplied source code, and we show its visualization on Figure 12.

---

**Algorithm 2** MKC($\mathcal{C}$) - 1 head calibration of thresholds for all possible $k$

---

**Require:** $\mathcal{C}$ (Calibration set of inputs)
1: $\Theta_r = \emptyset, \forall r$ {empty sets of observed interval sequences}
2: **for** $X \in \mathcal{C}$ **do**
3:    **if** is_prefill($X$) **then**
4:       $\boldsymbol{A} = Q(X)K^T(X)$
5:       $n =$NumRows($\boldsymbol{A}$)
6:       **for** $r = 1$ to $n - 1$ **do**
7:          $\boldsymbol{v} =$Sort($\boldsymbol{A}_r$)
8:          $\Theta_n = \Theta_n \cup \{\langle r - 1, [v_0, v_1]\rangle, \langle r - 2, [v_1, v_2]\rangle, \dots, \langle 1, [v_{r-2}, v_{r-1}]\rangle\}$
9:       **end for**
10:    **else** {generative decoding, $X \in \mathbb{R}^d$}
11:       $\boldsymbol{a} = Q(X)K^T(X)$
12:       $n =$Length($\boldsymbol{a}$)
13:       $\boldsymbol{v} =$Sort($\boldsymbol{a}$)
14:       $\Theta_n = \Theta_n \cup \{\langle n - 1, [v_0, v_1]\rangle, \langle n - 2, [v_1, v_2]\rangle, \dots, \langle 1, [v_{n-2}, v_{n-1}]\rangle\}$
15:    **end if**
16: **end for**
17: $\bar{k}_r(\theta) =$MergeIntervals($\Theta_r$), $\forall r$
18: **Return** $\theta_r(k) = \bar{k}_r^{-1}(k), \forall r$

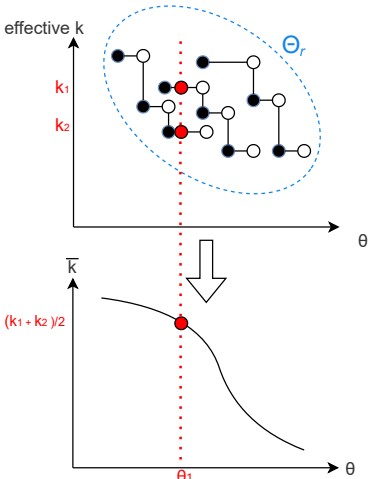

Figure 12: **MergeIntervals**($\Theta_r$) subroutine is merging the set $\Theta_r$ of interval sequences by averaging effective $k$ of all overlapping intervals across all interval sequences in the set $\Theta_r$.

Figure 13 illustrates the distribution of merged intervals ($\bar{k}(\theta)$) for selected attention layers and heads in the LLaMA2-7B model, following calibration via MKC on the Hellaswag dataset.

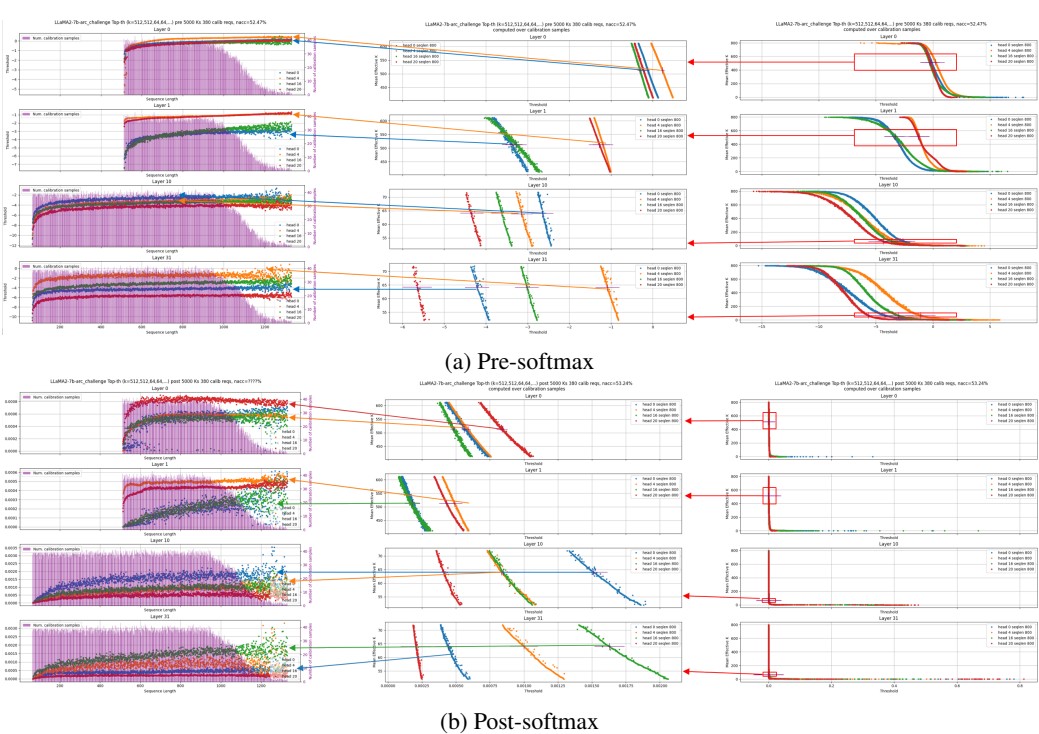

(a) Pre-softmax

(b) Post-softmax

Figure 13: MKC calibration of LLaMA2-7b on Hellaswag dataset. Right: the $\bar{k}_r(\theta)$ function as obtained from Algorithm 2 by merging the intervals for sequence length $r = 800$. Middle: zoom into a range of interest of $\bar{k}_r^{-1}(k)$ around $k = 64$, the selected threshold is marked on using $\theta_r(64) = \bar{k}_r^{-1}(64)$, Left: thresholds obtained from MKC for all sequence lengths (also other than $r = 800$.

It is worth mentioning that the main downside of the multi-k calibration is that it is very time and memory-consuming to perform all these interval mergings individually for every layer, head, and attention row id (i.e., sequence length). Speeding up the MergeIntervals routine, using reasonable approximations (such as subsampling) is an interesting research direction. The second slight downside is that it does not allow to apply $\text{top}_k$ at calibration, as we used to apply in Algorithm 1, since during MKC there is a multitude of potential $k$ parameters we are targeting.

## G   V-Mean Compensation

In this section, we provide a formal proof for our $\boldsymbol{V}$-Mean Compensation (VMC) serving as a good approximation of the sparsified $\tilde{\boldsymbol{s}}\tilde{\boldsymbol{V}}$ product to the full $\boldsymbol{s}\boldsymbol{V}$ product.

**Lemma G.1.** *Let $\boldsymbol{s} \in [0,1]^n$ denote the post-softmax attention vector, and let $\tilde{\boldsymbol{s}} \in [0,1]^{\tilde{k}}$ denote its thresholded variant, containing the $\tilde{k}$ selected values of $\boldsymbol{s}$, and let $\mathcal{I} \in \{0,\ldots,n-1\}^{\tilde{k}}$ denote the indices of $\boldsymbol{s}$ that surpassed the threshold such that $\forall i \in \{0,\ldots,\tilde{k}-1\} : \tilde{\boldsymbol{s}}_i = \boldsymbol{s}_{\mathcal{I}_i}$. Let $\boldsymbol{V} \in \mathbb{R}^{n \times d}$ be the value matrix and let $\tilde{\boldsymbol{V}} \in \mathbb{R}^{\tilde{k} \times d}$ be consisting only of the selected $\boldsymbol{V}$ rows such that $\forall i \in \{0,\ldots,\tilde{k}-1\} : \tilde{\boldsymbol{V}}_i = \boldsymbol{V}_{\mathcal{I}_i}$. We claim that in the expectation, the full product $\boldsymbol{p} = \boldsymbol{s}\boldsymbol{V} \in \mathbb{R}^d$ is equal to the thresholded attention plus the residual probability mass $\beta$ multiplied by the mean $\boldsymbol{V}$ row $\boldsymbol{\mu}$ (Equation (5)). Namely,*

$$\forall 0 \le j < d : E[\boldsymbol{p}_j] = (\tilde{\boldsymbol{s}}\tilde{\boldsymbol{V}})_j + \beta\boldsymbol{\mu}_j \tag{7}$$

*Proof.*

$$
\begin{aligned}
E[\boldsymbol{p}_j] &= E\Big[\sum_{i=0}^{n-1} s_i \boldsymbol{V}_{ij}\Big] \\
&= E\Big[\sum_{i\in\mathcal{I}} s_i \boldsymbol{V}_{ij}\Big] + E\Big[\sum_{i\in\bar{\mathcal{I}}} s_i V_{ij}\Big] \\
&= (\tilde{\boldsymbol{s}}\tilde{\boldsymbol{V}})_j + \sum_{i\in\bar{\mathcal{I}}} E[s_i \boldsymbol{V}_{ij}] \\
&\underset{s \perp \boldsymbol{V}_j}{=} (\tilde{\boldsymbol{s}}\tilde{\boldsymbol{V}})_j + \sum_{i\in\bar{\mathcal{I}}} E[s_i]E[\boldsymbol{V}_{ij}] \\
&\underset{s,\boldsymbol{V}_j \text{ uniform}}{=} (\tilde{\boldsymbol{s}}\tilde{\boldsymbol{V}})_j + \sum_{i\in\bar{\mathcal{I}}} \frac{\beta}{n-\tilde{k}}\boldsymbol{\mu}_j \\
&= (\tilde{\boldsymbol{s}}\tilde{\boldsymbol{V}})_j + (n-\tilde{k})\frac{\beta}{n-\tilde{k}}\boldsymbol{\mu}_j \\
&= (\tilde{\boldsymbol{s}}\tilde{\boldsymbol{V}})_j + \beta\boldsymbol{\mu}_j
\end{aligned}
\tag{8}
$$

$\square$

Under the following assumptions:

1. $s \perp \boldsymbol{V}_j$, that is the attention vector $\boldsymbol{s}$ is statistically independent on the elements in the columns of matrix $\boldsymbol{V}$. They are conditionally independent given the input $X$ from which they were originally computed via $\boldsymbol{V} = \boldsymbol{X}\boldsymbol{W}_{\boldsymbol{V}}$.

2. The distribution of $s_i, \forall i \in \bar{\mathcal{I}}$ within the long tail of the non-selected indices is close to uniform, and hence we can approximate its expectation by an average.

3. The expectation of $V_{ij}$ can be approximated by its average.

## H   Evaluation statistics

In this section, we present again the experimental results from Section 4.1; however, to demonstrate statistical significance, we show the error bars. This is important since every data point is aggregated using averaging across layers, heads, and test examples, as we will describe below. Therefore standard deviation of such an averaged metric is of interest.

Figure 14 focus on Q& tasks, showing the tradeoff between the model's accuracy (y-axis) and the number of attention elements as selected by the Top-k or Top-$\theta$ (x-axis). The standard deviation in the accuracy was provided by the LM-Eval evaluation harness, as a part of the standardized evaluation procedure. The average ratio presented on the x-axis and its standard deviation were computed over

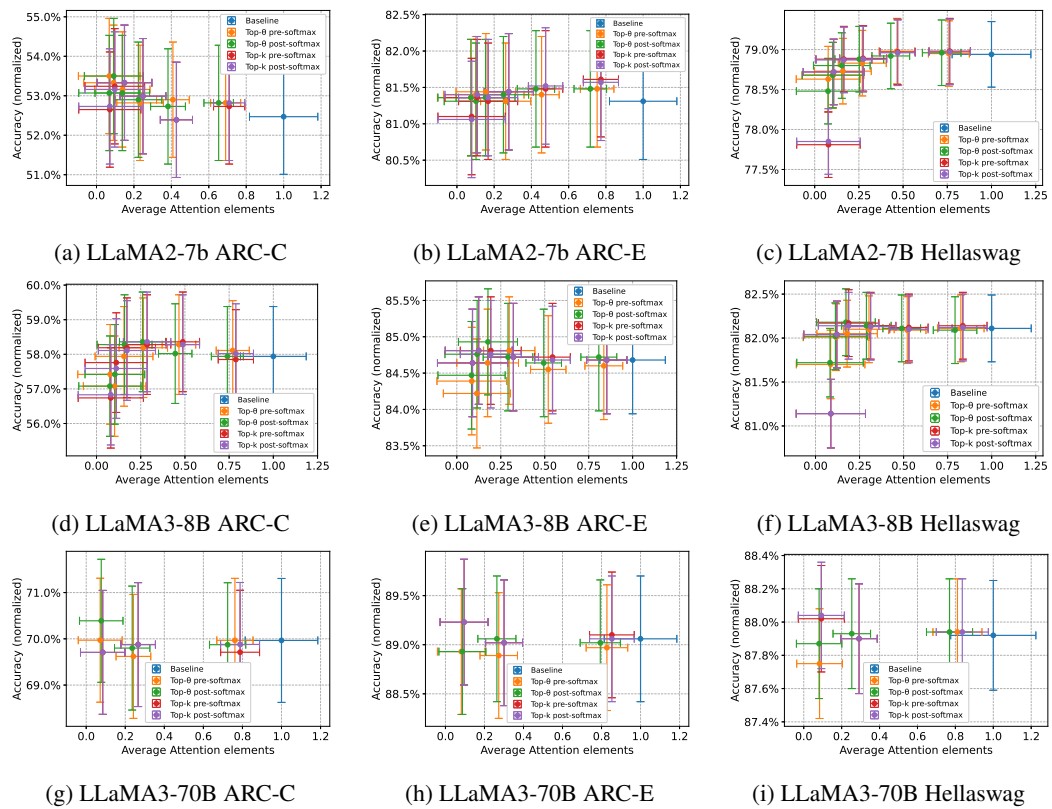

(a) LLaMA2-7b ARC-C  (b) LLaMA2-7b ARC-E  (c) LLaMA2-7B Hellaswag

(d) LLaMA3-8B ARC-C  (e) LLaMA3-8B ARC-E  (f) LLaMA3-8B Hellaswag

(g) LLaMA3-70B ARC-C  (h) LLaMA3-70B ARC-E  (i) LLaMA3-70B Hellaswag

Figure 14: **Prefill-based tasks** - Tradeoff between model accuracy averaged across test samples (y-axis), and the portion of kept attention elements per attention head (x-axis). All post-softmax Top-k and Top-θ employ VMC, and all pre-softmax variants employ both VMC and exact SDC. Using the VMC and the SDC compensations achieves little if any accuracy degradation while achieving up to 10× reduction in the attention elements.

the following population: number of test samples × number of model layers × number of attention heads. To present the ratios, we first compute the absolute average and absolute standard deviation of the total count of attention elements and in the attention matrix, second - we normalize both the standard deviation and the average by the average number of attention elements when no sparsification took place.

Figures 15 and 16 show the evaluation on Human-eval dataset. Figure 15 is similar to the Q&A plots as it shows how the reduction in the number of attention elements during *prefill phase* impacts the accuracy score (pass@1). Figure 16 focuses on the generative *decoding phase* and shows how the number of the needed $V$ rows is affected by the elements that were selected in every row independently, which introduces the effect of GQA - since the indices of the selected elements are united across the heads in the group. The average and the standard deviation in this plot are taken across the following population of samples: number of test tasks × number of autoregressive forward passes × number of model layers × number of attention heads ×.

## I  PRE- VS. POST-SOFTMAX SPARSIFICATION

This section provides extended experiment plots comparing pure pre-softmax and post-softmax accuracy on Q&A tasks. In all of them the post softmax consistently achieves higher scores.

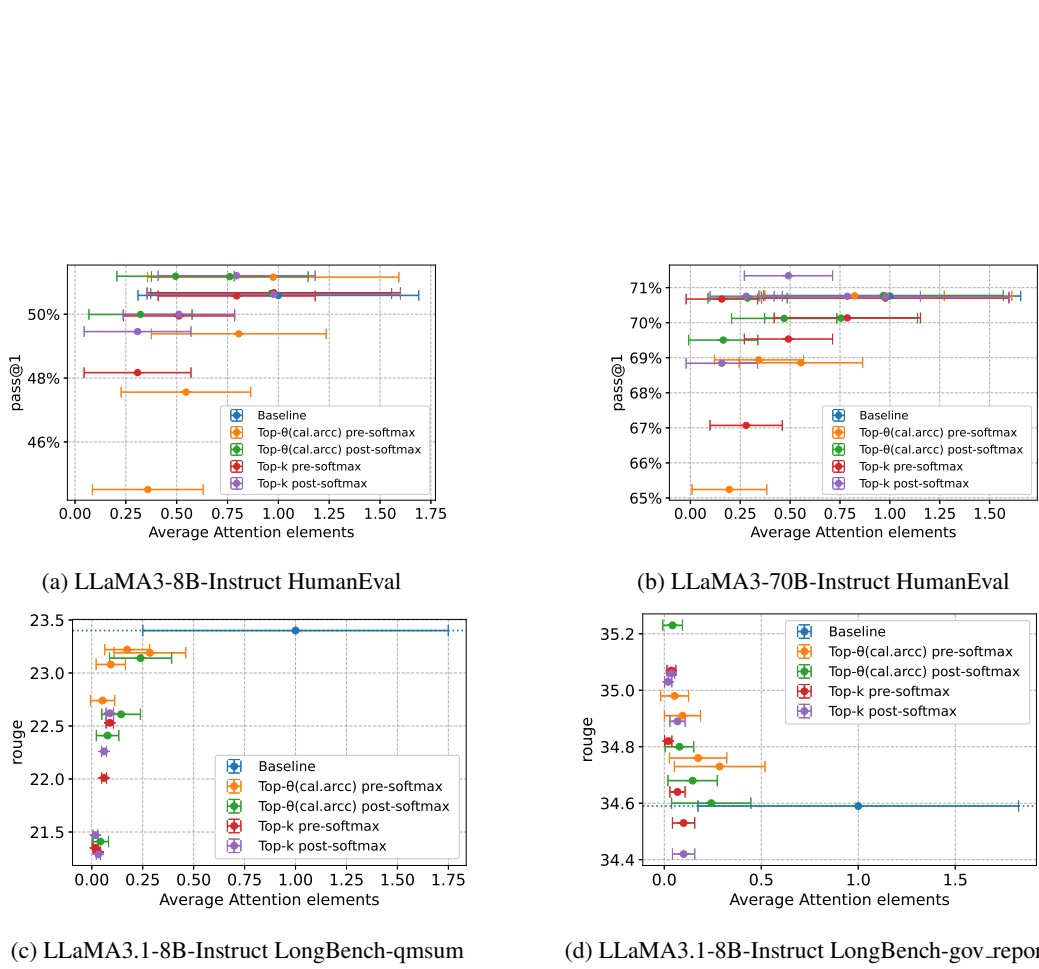

(a) LLaMA3-8B-Instruct HumanEval

(b) LLaMA3-70B-Instruct HumanEval

(c) LLaMA3.1-8B-Instruct LongBench-qmsum

(d) LLaMA3.1-8B-Instruct LongBench-gov_report

Figure 15: **Generative Tasks - during prefill** - Tradeoff between model accuracy averaged across test samples (y-axis), and the portion of required attention elements per head (x-axis). The Top-$\theta$ variants employ threshold calibrated on ARC-C dataset. All post-softmax Top-k and Top-$\theta$ employ VMC, all pre-softmax variants employ both VMC and exact SDC

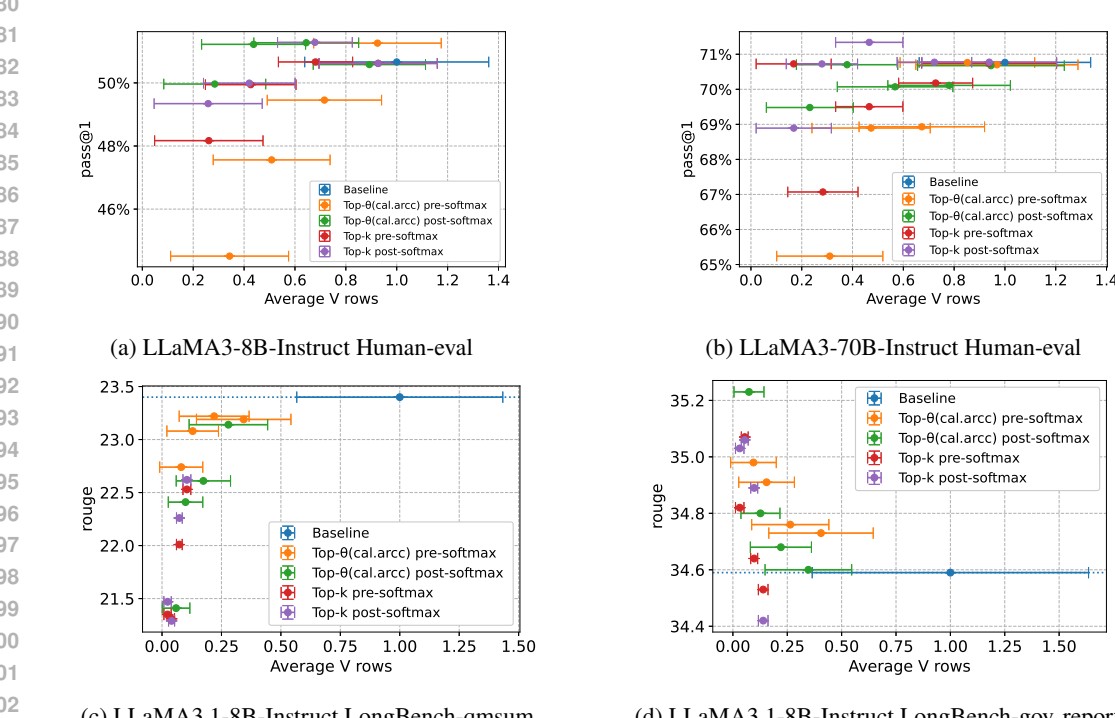

(a) LLaMA3-8B-Instruct Human-eval

(b) LLaMA3-70B-Instruct Human-eval

(c) LLaMA3.1-8B-Instruct LongBench-qmsum

(d) LLaMA3.1-8B-Instruct LongBench-gov_report

Figure 16: **Generative Tasks - during generative decoding** – Tradeoff between model accuracy averaged across test samples (y-axis) and the portion of required $V$-rows per group of heads (x-axis). The Top-$\theta$ variants employ threshold calibrated on ARC-C dataset. All post-softmax Top-k and Top-$\theta$ employ VMC, and all pre-softmax variants employ VMC and exact SDC. 3× to 10× reduction of $V$ rows is achieved.

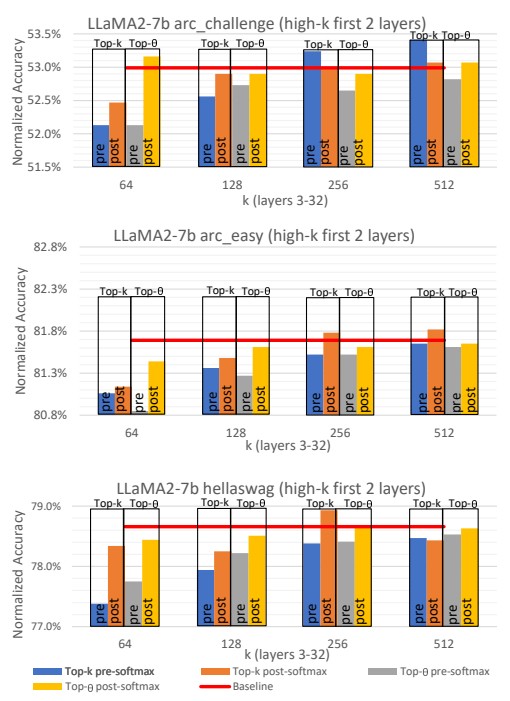

Figure 17: Comparison of Pre- and post-softmax thresholding

## J    THRESHOLDING DIFFERENT LAYERS

In this appendix, we present more results that support the decision to use denser initial layers. That is to use higher $k$ for Top-k or calibrating towards a higher $k$ in Top-θ. Figure 18 shows LLaMA2-7b model accuracy on Q&A datasets, and LLaMA2-70b model accuracy for Hellaswag dataset. The figure compares Top-k and Top-θ using higher $k$ in the first 2 layers against using equal $k$ in all layers. All variants do not perform any compensations.

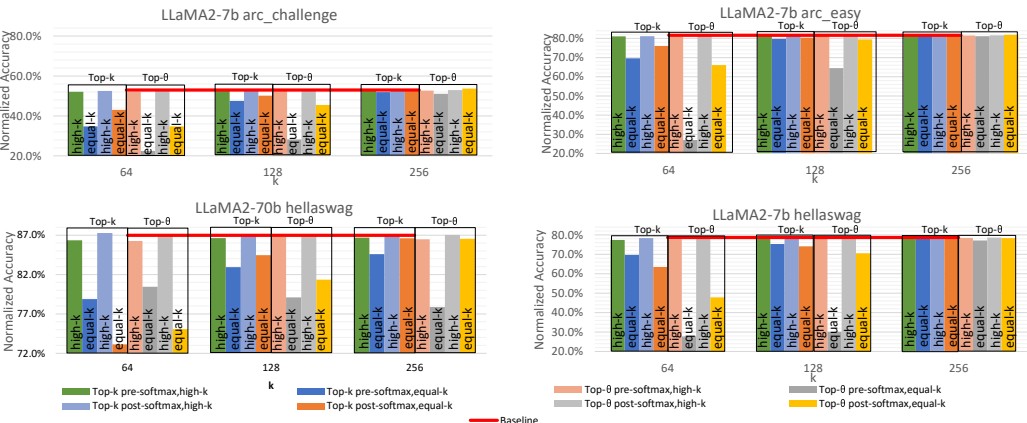

Figure 18: The positive impact of keeping first two layers dense (higher k for calibration), compared to keeping equal k in all layers

## K    THRESHOLDING DIFFERENT ATTENTION ROWS

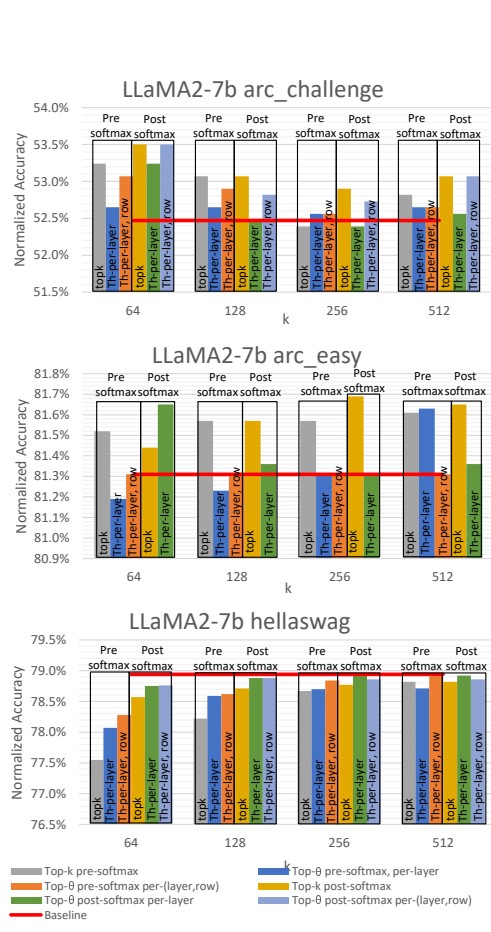

Figure 19: Calibrating per-attention-row thresholds vs a unified threshold for all rows (sequence lengths)

## L  IMPACT OF GQA

In Figure 20, we show how the different attention sparsification approaches (Top-k, Top-$\theta$ with and without CAPK) affect the number of required $V$ rows. Top-k ( Figure 20a) guarantees exactly 128 selected elements per row of every head. However, the unified set of 4 heads in the group reaches only about 250, which indicates a certain agreement between the heads, which is mostly found in the recent tokens (as seen on the heatmap). We observed very similar characteristics in other heads and layers. Top-$\theta$ approach with capping the number of selected elements to at most 128 per head yielded degraded quality of the generated text since it mainly focused attention on the most recent tokens. Finally, the ordinary Top-$\theta$ in Figure 20c, which provided good quality results, does seem to exhibit a certain variability in the number of selected elements per group, sometimes selecting more than 128 per group.

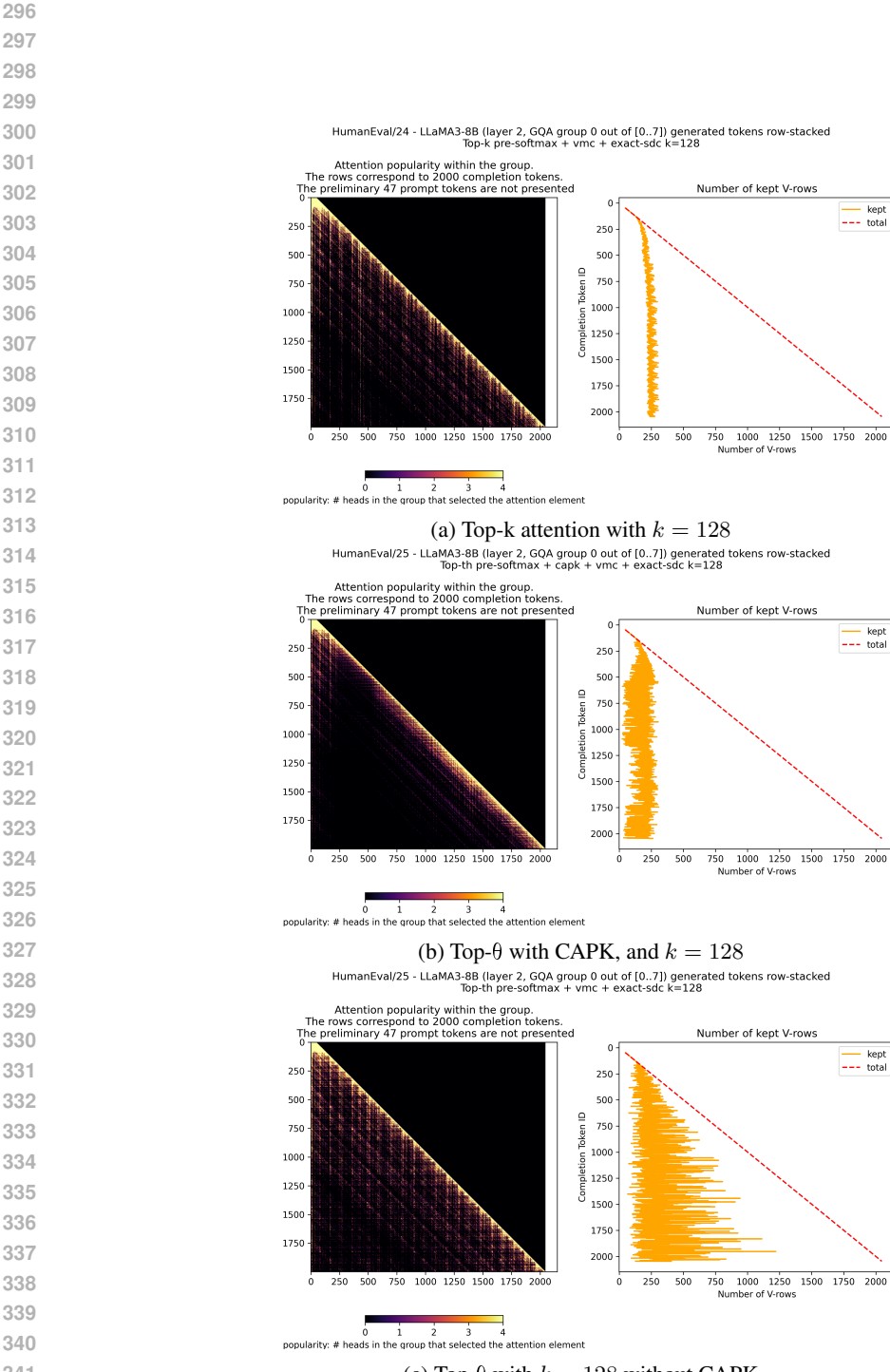

(a) Top-k attention with $k = 128$

(b) Top-$\theta$ with CAPK, and $k = 128$

(c) Top-$\theta$ with $k = 128$ without CAPK

Figure 20: **Attention popularity mask** – LLaMA-3-8B (GQA group size$= 4$), Human-eval task number 25, generative decoding iterations as rows. Left – heat map showing how many heads had the corresponding attention element in their Top-128; on the right – the number of $\boldsymbol{V}$-rows required to be used (1 head in the group is enough to require a $\boldsymbol{V}$ row).

