# OpenReview forum: "Top-Theta Attention: Sparsifying Transformers by Compensated Thresholding"
_ICLR.cc/2026/Conference — Submitted to ICLR 2026_

### Official Review · Reviewer_zNgo · 2025-10-23

**Soundness:** 2
**Presentation:** 2
**Contribution:** 2
**Rating:** 2
**Confidence:** 4

**Summary:**

This paper proposes a novel method called Top-$\theta$ attention, which can approximate top-$k$ attention computation without using sorting algorithms or other approximation variants. This approach demonstrates a 3- to 10-fold reduction in V-cache usage and up to 10 times fewer attention elements during inference, while degrading accuracy by no more than 1%.

**Strengths:**

* **High Efficiency**: Reduces **V-cache usage by 3–10×** and attention elements by **up to 10×** with <1% accuracy loss.

* **Robust to Domain Shift**: Thresholds are **model-intrinsic** — calibrated once and work across tasks and datasets.

* **Better Than Top-\(k\)**: Replaces expensive top-\(k\) search with **constant-time thresholding**, removing row-wise dependencies.

**Weaknesses:**

* This paper lacks a comprehensive review of the field of sparse attention, which has already been popular to study for a long time.
* This paper lacks a comprehensive experimental comparison with current popular methods; it is not clear how this method outperforms other sparsity-based attention approximation algorithms.
* The authors present a new method with promising performance in this work, but after checking the whole paper, it is still not clear what this paper wants to solve. If they just provide a new method that approximates attention efficiently without losing too much accuracy, they should do a comprehensive experimental comparison with current popular methods; otherwise, the community cannot tell why this work is better than others and why people should use this method rather than others.
* This paper lacks a comparison of the efficiency among the top-$\theta$ attention and other methods, especially when it involves additional inference; the additional computational cost should be considered, too.

**Questions:**

* Please see Weaknesses
* Line 124-125 "We conjecture that the top-k search algorithms....": Could you explain why this conjecture holds?

---

> ### Author Response · Authors · 2025-11-20
>
> Thank you for your evaluation and the detailed feedback. We appreciate your recognition of our method's efficiency, robustness to domain shift, and its theoretical advantage over standard Top-$k$ search. We have carefully considered your concerns and rewritten significant parts of the manuscript. Below, we outline the specific revisions made to the manuscript to address these points.
>
> **1. Comprehensive Review and Comparison (Addressing Weaknesses 1-3)**
> We acknowledge that the initial submission did not sufficiently position Top-$\theta$ within the rapidly evolving landscape of sparse attention.
> - Revised Related Work (Section 5): We have fundamentally redone the Related Work section to include a thorough review of state-of-the-art methods from 2024 and 2025.
> - Comparative Table: We have added a detailed table 1 qualitatively comparing Top-$\theta$ against these recent approaches. This table contrasts key attributes such as the selection mechanism (e.g., Sorting vs. Estimation vs. Thresholding), computational cost, and calibration requirements.
> - Problem Clarity and Motivation: We have introduced a new Section 2.4 to give a clear motivating example of paged-attention, where Top-$\theta$ unblocks the global all-pages-wide synchronization inefficiency of the sorting operations required by Top-$k$ and similar methods.
>
> **2. Efficiency Analysis (Addressing Weakness 4)**
> Regarding the concern about additional inference costs:
> - We added a new kernel top-theta kernel experiment as Section 4.7
> - We have clarified in the methodology and experiments that the computational overhead of Top-$\theta$ is negligible compared to the baselines.
> - While Top-$k$ requires identifying the $k$-th largest element (often $O(N \log k)$ or requiring synchronization primitives), our method relies on a constant-time threshold check per element ($O(1)$ parallelized), which is significantly faster and more hardware-friendly.
>
> **3. Verification of Conjecture (Addressing Question regarding Lines 124-125).**
> The conjecture posits that a static, calibrated threshold $\theta$ results in retaining the desired number of tokens $k$ on average, without enforcing a hard cap per row. We verified this empirically early in our study. Please refer to Figure 10 in the manuscript (Calibration Validation), where the Y-axis displays the ratio between the number of tokens selected by Top-$\theta$ and the target $k$. As illustrated, with a moderately sized calibration set (>100 samples), this ratio converges tightly to $1.0$ with very low variance. This confirms that our calibrated threshold effectively acts as a statistically valid, constant-time proxy for the expensive Top-$k$ selection.We believe these revisions provide the comprehensive context and rigorous comparison required.
>
> Thank you for the opportunity to strengthen our work.

---

> > ### Comment · Reviewer_zNgo · 2025-11-27
> >
> > Hi, thanks for your reply. As you have already solved most of my concerns, I would like to raise my score to 4. The reason why I keep the score below the acceptance line is that even though this work provides an interesting idea, the insights and contributions that it will bring to the community are still unclear. I suggest that you compare your work with not only simple baselines, but also with current SOTA methods to evaluate your performance. Moreover, reconstructing the first two sections for a better and professional presentation, clearly introducing the key problem the community cares about, and the corresponding important results that this work has achieved, will be an excellent way to improve this work.

---

### Official Review · Reviewer_WyYY · 2025-10-31

**Soundness:** 2
**Presentation:** 2
**Contribution:** 1
**Rating:** 2
**Confidence:** 5

**Summary:**

This paper proposes Top-Theta Attention, a training-free method for sparsifying transformer attention during inference. The key idea is that static, per-head thresholds can be calibrated to retain the desired constant number of significant elements per attention row.

**Strengths:**

+ The proposed method does not require training.
+ The proposed method shows effectiveness on the state-of-the-art open-source LLM models.

**Weaknesses:**

- Limited novelty. Sparse attention has been extensively studied, and many state-of-the-art methods already exist. The proposed Top-Theta or threshold-based sparsification appears to be a minor variation of the well-known top-k attention, which has been applied both in standard transformer architectures and in large language models. The contribution, therefore, seems incremental rather than fundamentally new.

- Unclear motivation. The paper does not clearly explain why a content-based attention mechanism is necessary or how it addresses specific limitations of existing methods. The proposed approach essentially performs pruning by comparing attention scores against a threshold, but it remains unclear why this thresholding scheme is conceptually or practically superior.

- Unjustified design choices and missing ablations. Several components of the proposed method are introduced without sufficient justification or explanation. Moreover, the paper lacks ablation studies to demonstrate the contribution of each component or to support the design decisions made.

- Insufficient experimental validation. The experimental section lacks comprehensive comparisons with existing sparse-attention methods and does not include evaluations on modern large-scale LLMs. As a result, it is challenging to evaluate the generality and practical effectiveness of the proposed technique.

**Questions:**

See the Weaknesses.

---

> ### Author Response · Authors · 2025-11-19
>
> Thank you for your honest and professional feedback, thanks to which we have gladly **added an important motivating paged-attention example in a new Section 2.4**. We appreciate the opportunity to clarify and highlight the significant contributions of our work. We would like to use this opportunity to address the weaknesses you have pointed out:
>
> **1. Limited Novelty**
>
> Our primary contribution is showing that *model-intrinsic attention distribution statistics enable effective static per-head, per-row thresholds for sparsification*, which can replace the expensive top-k operations with simple thresholding. Unlike prior art that focuses on dynamic top-k selection [Gupta et al., 2021], we demonstrate that *a one-time calibrated static threshold suffices across datasets and tasks*, including large models up to 70B parameters. This insight is crucial for enabling tiled and pipelined attention kernels used in practical decoders, particularly for paged attention in vLLM-like infrastructures.
>
> The new Section 2.4 we have now added provides a motivating example showing that Top-theta directly addresses the **fundamental architectural bottleneck** where modern flash attention kernels, in particular paged-attention [Kwon et al. 2023], process KV pages sequentially (each page undergoes QKᵀ then Softmax then SV - without a global barrier after the Softmax). This efficient pipelined execution, however, makes it infeasible to perform top-k for each page separately since the global ranking of the attention cannot be determined before SV without a global barrier. Our thresholding scheme doesn't have the full-row requirement, allowing early pruning within each page, supporting **low-latency, pipelined decoding kernels**, as described in “FlashAttention” and subsequent research [Dao et al., 2022, 2023]. In our humble opinion, this distinction in algorithmic design and its hardware implications represents a substantial innovation.
>
> **2. Unclear Motivation**
>
> A dedicated Section 2.4 should now clarify that Top-theta attention is both practical and superior to the top-k, based on the widely used paged-attention kernel. Because the full softmax for each KV page is computed independently and sequentially, the ability to apply early thresholding with static thresholds is critical to enabling efficient tiling and distributed inference without waiting for global top-k computations. This solves a practical problem increasingly relevant as sequences scale to tens of thousands of tokens.
>
> **3. Unjustified Design Choices and Missing Ablations**
>
> Kindly please refer to the outline of the ablations we have included in the paper:
>
> - Sec. 3.1: Justifies the (1) per-head and (2) per-row individual threshold calibration that are important in top-theta design.
> - Sec. 4.2: Pre-softmax vs post-softmax thresholding impact - justifying the design decisions referred to the placement of the thresholding in the attention kernel.
> - Sec. 4.3: Justification to apply lower sparsification pressure on the initial layers in the model.
> - Sec. 4.4: Ablation of numerical compensation methods - SDC & VMC - their variants and combinations.
> - Sec. 4.5: Effect of GQA - how grouped heads' token selections impact data movement.
> - Sec. 4.6: Distribution shift resilience - justifies the key "calibrate once per model" attribute of our work.
>
> In our humble opinion, these ablations demonstrate the careful and principled design underlying each component.
>
> **4. Insufficient Experimental Validation**
>
> The paper presents **extensive evaluations on modern large language models**, including LLaMA-3-70B, covering diverse tasks such as Q&A, code generation (HumanEval), and long-context summarization (LongBench). We show generality by demonstrating resilience to distribution shifts. Our results on these flagship models with up to 70B parameters clearly establish the scalability and generality of the approach, and show Top-theta sparsification matches or improves over dense baselines with a fraction of the attention elements.
>
> Due to the conceptual focus of this paper, we have not exhaustively compared to all sparse attention variants from the literature, but intentionally highlighted the core intrinsic property of threshold-based sparsifiability of LLMs. Given the ICLR emphasis on fundamental advances in efficient representations, we believe our paper offers valuable new insights to the community.
>
> **We thank you again for providing impactful feedback that reshaped our paper, and kindly ask you to reconsider the scientific merit of our work in light of these clarifications.**
>
>
> ### References
>
> - Gupta et al., Memory-Efficient Transformers via Top-k Attention, 2021
> - Dao et al., FlashAttention: Fast and Memory-Efficient Exact Attention with IO-Awareness, 2022
> - Dao et al., Flashattention-2: Faster attention with better parallelism and work partitioning, 2023
> - Kwon et al., Efficient memory management for large language model serving with pagedattention, 2023

---

> > ### Comment · Reviewer_WyYY · 2025-11-28
> >
> > Thank you for the response. I would like to increase my rating to 4. However, from my perspective, the contribution is still incremental to existing methods, and it cannot exceed 4.

---

### Official Review · Reviewer_PArU · 2025-11-02

**Soundness:** 4
**Presentation:** 4
**Contribution:** 3
**Rating:** 8
**Confidence:** 4

**Summary:**

The paper proposes Top-Theta Attention, a training-free sparsification method for transformer attention at inference. Key idea: instead of doing per-row top-k search, the authors pre-calibrate a static per-head, per-row threshold so that, on average, each attention row keeps ≈k important entries. At inference, attention scores are just compared to this threshold — a pure elementwise op — so no row-wise dependency, and it works with tiled / distributed kernels. To offset the loss from dropping entries, they add two compensations: Softmax Denominator Compensation (SDC) to better approximate post-softmax sparsity, and V-Mean Compensation (VMC) to add back the average contribution of discarded tokens. On LLaMA-2, LLaMA-3 and LLaMA-3.1 (7B–70B), across ARC-C/E, HellaSwag, HumanEval, and LongBench, they report 3–10× fewer V-rows / attention elements with ≤1% accuracy loss, sometimes even small gains, and the thresholds calibrated on ARC-C transfer to code and long-context tasks, suggesting the thresholds are model-not-data specific. Calibration needs only a few hundred samples and is one-time per model.

**Strengths:**

1. **Training-free & model-centric.** Calibrate once per model, a few hundred samples, then reuse across domains (ARC-C → HumanEval/LongBench) — this is much cheaper than retrain/fine-tune-based sparsity.
2. **Tile- and kernel-friendly formulation.** Pure elementwise thresholding; no per-row top-k that breaks tiling. This is exactly where existing top-k attention hurts.
3. **Strong empirical coverage.** LLaMA-2, LLaMA-3, LLaMA-3.1; 7B→70B; prefill (QA) and decoding (HumanEval, LongBench); GQA case analyzed.
4. **Storage overhead negligible**. Thresholds are tiny (~12MB for LLaMA-3-70B).

**Weaknesses:**

1. **No wall-clock / kernel-level evaluation.** The main selling point is “better for tiled / distributed kernels than top-k,” but the paper does not show an actual implementation or runtime vs. FlashAttention+Top-k baselines on GPU.
2. **Calibration cost vs. model/library changes not fully discussed.** If KV layout or rotary settings change (common in serving stacks), do we need to recalibrate?

**Questions:**

1. **On SDC choice.** You present three SDC estimators (offline, exp-threshold, exact). For an actual GPU kernel, which one do you expect to be used, and what’s the real compute/memory overhead relative to plain Top-j? Please quantify for LLaMA-3-8B decode (one token).
2. **GQA union strategy.** You mention two alternatives (discard low-support V-rows; or union across heads). Did you test either? If not, what made you decide on per-head selections only?
3. **Per-row vs parametric thresholds.** You mention fitting a parametric function to rows to reduce storage. Did you actually try it, and what is the accuracy drop vs. full per-row table?

---

> ### Author Response · Authors · 2025-11-19
>
> Thank you for the thorough and constructive review. We deeply appreciate your detailed engagement with our work, and we have made our best to incorporate as many of your comments into the revised manuscript. Below, we address each of your questions systematically.
>
> 1. **SDC** that is the most practical is the _exp-threshold_ since it requires $3\cdot1024$ arithmetic-scalar operations and no row dependence and no calibration, and no additional memory, whereas in terms of accuracy, it performed similarly well to the exact. The _calibrated_ SDC is slightly less practical as it requires its own calibration, requires adding the same memory as for the thresholds themselves (although small, 2.5MB for the entire LLaMA3-8B), yet needs only $1\cdot1024$ scalar-load operations to retrieve the calibrated factors. The most expensive is the _exact_ SDC, having the cost of $2(n-\tilde{k})\cdot 1024$ extra scalar operations to compute the missing part of the denominator. Note that the 1024 factor corresponds to the number of layers multiplied by the number of heads in LLaMA3-8B. Having all that said, since the generative decoding of MHSA and GQA models is memory-bandwidth-bound, all these compute costs are not expected to become bottlenecks.
> 2. **GQA treatment**, we chose the most canonical way of supplying each attention head with its chosen V-entries since this is the approach that does not involve any extra mask processing and hence is the most streamlined for implementing it in a kernel. The alternative approach of "discarding V-rows requested by only a few heads" introduces more turning knobs (e.g. minimum number of heads to vote for an index to be kept) as well as risks dropping crucial tokens, whereas the "union-across-heads" approach is definitely more straightforward to implement, which we tried very briefly and saw insignificant uplift in accuracy compared to the canonical way we sticked to eventually.
> 3. **Parametric thresholds** were not tried out by us and left for a separate study. We do know confidently that per-row threshold tables exhibit very smooth curves; therefore, a good fit should not result in an accuracy degradation. Secondly, the fitted solution might not be the most practical one for a high-performance kernel, because a simple lookup of a single threshold in a table is probably faster than loading a few parameters from the same memory and computing the threshold using a fitted function. Lastly, determining the threshold value (per head) takes place once per forward pass of a layer, hence the latency of either table-lookup or fitted-function evaluation becomes negligible compared to attention layer execution time. Nevertheless, threshold parameterization is a valid question to make the Top-theta scalable, especially for extremely long sequence scenarios.
>
> Regarding the weaknesses:
> 1. **Wall/kernel clock time** - We have implemented a prototype of the Top-theta MHSA attention kernel on Ascend NPU and observed a kernel speedup of $1.17\times$ scompared to an optimized flash attention kernel, when the target $k$ is at least 128 out of $n=2048$ tokens. However, such kernel-level performance gains are extremely hardware-dependent, involving low-level specifications of DMAs, bandwidth, and parallel execution capabilities. Thus, throughout our paper, we keep an algorithmic focus to maintain wide applicability.
> 2. **Calibration cost vs. library changes** - Calibration remains a lightweight one-time offline process requiring only a few hundred samples. Thresholds primarily reflect inherent model attention distributions, so routine changes like KV layout or rotary embedding adjustments typically do not require recalibration. If drastic architectural changes occur, recalibration is a fast and practical alternative to costly retraining or fine-tuning.

---

### Author Response · Authors · 2025-12-03
**Rebuttal Summary For AC**

This paper has one high-confidence accept (8) that highlights its clear technical soundness, excellent presentation, and practical impact: a training-free, model-centric sparsification scheme that replaces row-wise top-k with static, per-head thresholds, enabling tiled and paged attention kernels to achieve 3-10x reductions in V-cache and attention elements on LLaMA-2/3/3.1 (7B-70B) with <1% accuracy loss.

The rebuttal led both initially negative reviewers to raise their scores to 4 after the authors substantially strengthened the motivation (tiled top-k attention bottlenecks), clarified the conceptual novelty (model-intrinsic calibrated thresholds as a hardware-friendly alternative to top-k), expanded ablations and related-work positioning, and added a wall-clock kernel-level experiment to quantify the actual speedup gains.

Given this clearly positive review trajectory, the absence of unresolved correctness concerns, and the strong alignment with ICLR’s goals on efficient representation and processing (capitalizing on sparsity), we would like to advocate that it would be beneficial to the community to accept this work as a poster.

---

### Meta-Review · Area_Chair_4F4Q · 2025-12-26

**Summary:**

The paper received mixed reviews, consisting of one positive and two negative ratings. Although the two negative reviewers engaged in the discussion following the rebuttal and increased their scores, they maintained concerns that the authors failed to address the comparison against current state-of-the-art (SOTA) methods. Additionally, the reviewers suggested that the authors improve the presentation to better illustrate the problem studied and the results achieved. Overall, because the contribution of the current version is incremental, the paper is recommended for rejection.

**Reviewer Concerns:**

The authors addressed the concerns of design choices of the paper and showed the ablation study for demonstrating the contribution. They also provided better literature review with other works. However, the authors did not provide evidence showing the better performance compared with existing SOTA works.

**Reviewer Scores:**

Reviewer PArU has a positive score and the authors have provided rebuttal to answer the questions from reviewer.

Reviewer WyYY and zNgo both engaged in the discussion. However, their concerns about the comparison with existing works are not solved by the authors. The two reviewers might increase their ratings if the authors can provide the comparison with SOTA works.

---

### Decision · Program_Chairs · 2026-01-26

Reject